# Independence Testing for Temporal Data

**Cencheng Shen**                                                                *shenc@udel.edu*
*Department of Applied Economics and Statistics*
*University of Delaware*

**Jaewon Chung**                                                                  *j1c@jhu.edu*
*Department of Biomedical Engineering*
*Johns Hopkins University*

**Ronak Mehta**                                                                  *ronakdm@uw.edu*
*Department of Statistics*
*University of Washington*

**Ting Xu**                                                                *Ting.Xu@childmind.org*
*Child Mind Institute*

**Joshua T. Vogelstein**                                                            *jovo@jhu.edu*
*Department of Biomedical Engineering*
*Johns Hopkins University*

**Reviewed on OpenReview:** *https://openreview.net/forum?id=jv1aPQINc4*

## Abstract

Temporal data are increasingly prevalent in modern data science. A fundamental question is whether two time series are related or not. Existing approaches often have limitations, such as relying on parametric assumptions, detecting only linear associations, and requiring multiple tests and corrections. While many non-parametric and universally consistent dependence measures have recently been proposed, directly applying them to temporal data can inflate the p-value and result in an invalid test. To address these challenges, this paper introduces the temporal dependence statistic with block permutation to test independence between temporal data. Under proper assumptions, the proposed procedure is asymptotically valid and universally consistent for testing independence between stationary time series, and capable of estimating the optimal dependence lag that maximizes the dependence. Moreover, it is compatible with a rich family of distance and kernel based dependence measures, eliminates the need for multiple testing, and exhibits excellent testing power in various simulation settings.

## 1    Introduction

Temporal data, often referred to as time series, finds wide applications across diverse domains, such as functional magnetic resonance imaging (fMRI) in neuroscience, dynamic social networks in sociology, financial indices, etc. In a broader context, temporal data can be seen as a type of structural data characterized by inherent underlying patterns. When dealing with temporal data, a fundamental problem is to determine the presence of a relationship between two jointly observed time series.

In the context of standard independent and identically distributed (i.i.d.) data, where observations $(X_1, Y_1), (X_2, Y_2), \ldots, (X_n, Y_n)$ are drawn independently and identically from the joint distribution $F_{XY}$, the question simplifies to whether the underlying random variables $X$ and $Y$ are independent, i.e., $F_{XY} = F_X F_Y$. Many recent dependence measures have been proposed to tackle this problem, aiming to achieve valid and universally consistent independence testing. These methods include distance correlation (Szekely et al.,

2007; Szekely & Rizzo, 2009; 2014), Hilbert-Schmidt independence criterion (Gretton et al., 2005; Gretton & Gyorfi, 2010; Gretton et al., 2012), multiscale graph correlation (Vogelstein et al., 2019; Shen et al., 2020; Lee et al., 2019), and many others (Heller et al., 2013; Zhu et al., 2017; Pan et al., 2020).

However, the standard testing framework is not applicable to structured data such as time series, because the i.i.d. assumption often does not hold. As a result, standard testing procedures like the permutation test are known to produce inflated p-values and are thus unsuitable for testing structured data (Guillot & Rousset, 2013; DiCiccio & Romano, 2017). Existing research on testing independence for temporal data is limited, often relying on linear measures such as autocorrelation and cross-correlation, which may overlook potential nonlinear relationships (Wang et al., 2021). A commonly made assumption is to consider the sample data as stationary, meaning that the joint distribution of $(X_t, Y_{t-l})$ depends only on the lag $l$ and not on any specific time index $t$. Approaches for addressing the instantaneous time problem, where the goal is to detect whether $X_t$ and $Y_t$ are independent, have been explored in Chwialkowski & Gretton (2014). Moreover, Chwialkowski et al. (2014) investigates the problem of testing between $X_t$ and $Y_{t-l}$ for each lag $l$ separately, employing multiple testing techniques.

In this paper, we propose an aggregated temporal statistic and utilize a block permutation procedure to extend the scope of independence testing beyond the i.i.d. assumption. Given a standard dependence measure such as distance correlation, our method first calculates a set of cross dependence statistics. These statistics not only facilitate the estimation of the optimal dependence lag, but also enable the computation of the temporal dependence statistic as a weighted aggregation of all cross dependence statistics. Subsequently, we employ a block permutation procedure to derive a p-value for hypothesis testing. Under proper assumptions regarding the choice of the dependence measure, the joint distribution of the temporal data, and the parameters of the block permutation, we establish the asymptotic properties of the temporal dependence, and prove the asymptotic validity and universal consistency of our method. Notably, the proposed temporal dependence method is non-parametric and does not require multiple testing.

Numerically, we show that the proposed approach yields satisfactory testing power when applied to simulated time series with small sample sizes. It is compatible with various dependence measure choices, and numerically superior and more versatile than previously proposed time series testing procedures. Additionally, we present the results of two real-data experiments, utilizing the proposed method to analyze neural connectivity based on fMRI data, as well as uncovering interesting temporal dependencies between the general stock market and low-beta stocks.

## 2 Method

### 2.1 Hypothesis for Testing Temporal Dependence

Given the joint sample data $\{(X_1, Y_1), ..., (X_n, Y_n)\}$, let $\vec{X} = \{X_1, \ldots, X_n\} \in \mathbb{R}^{p \times n}$ and $\vec{Y} = \{Y_1, \ldots, Y_n\} \in \mathbb{R}^{q \times n}$ represent each individual sample data. Here, $p$ and $q$ denote the dimensions and are positive integers, and $n$ is the sample size.

Suppose $(\vec{X}, \vec{Y})$ is strictly stationary, meaning the distribution at any set of indices remains the same. We can represent the distributions of $X_t$ and $Y_t$ at any point $t$ as $F_X$ and $F_Y$, and represent the distribution of $(X_t, Y_{t-l})$ as $F_{XY_{-l}}$ for each lag $l \geq 0$.

We aim to test the following independence hypothesis between $\vec{X}$ and $\vec{Y}$:

$$H_0 : F_{XY_{-l}} = F_X F_Y \text{ for each } l \in \{0, 1, ..., L\}$$
$$H_A : F_{XY_{-l}} \neq F_X F_Y \text{ for some } l \in \{0, 1, ..., L\},$$

Here, $L$ is a non-negative integer denoting the maximum lag under consideration. Essentially, the null hypothesis states that $X_t$ is independent of present and past values of $Y_{t-l}$ for all of $l = 0, \ldots, L$. In contrast, the alternative hypothesis suggests $(\vec{X}, \vec{Y}_{-l})$ are dependent for at least one $l$ in the range of $[0, L]$.

This setting is, in fact, a generalization of the standard i.i.d. setting, where it was assumed that $(X_1, Y_1), (X_2, Y_2), \ldots, (X_n, Y_n) \overset{i.i.d.}{\sim} F_{XY}$, and the null hypothesis simplifies to $F_{XY} = F_X F_Y$ because there

is no possible dependence other than $l = 0$. Hence, our subsequent method and theory for testing two time series are also applicable when only one of them is time series or when both are standard i.i.d. data. Moreover, they are applicable to any general structured data that can be assumed stationary.

## 2.2 Main Algorithm

The proposed method consists of four steps: computation of the cross-lag dependence statistics, estimation of the optimal dependence lag, computation of temporal dependence statistic, and block permutation to obtain the p-value for testing purposes. Details regarding the choice of the dependence measure, block permutation, and computational complexity are discussed in the following subsections.

**Input:** Two jointly-sampled datasets represented as $\vec{X} \in \mathbb{R}^{p \times n}$ and $\vec{Y} \in \mathbb{R}^{q \times n}$, a given choice of sample dependence measure $\tau_n(\cdot, \cdot) : \mathbb{R}^{p \times n} \times \mathbb{R}^{q \times n} \to \mathbb{R}$, and three positive integers: the lag limit $L$, the number of blocks $B$, and the number of random permutations $R$.

**Step 1:** Compute the set of cross dependence sample statistics $\{\tau_n(\vec{X}, \vec{Y}_{-l}), l = 0, \dots, L\}$. Here, $(\vec{X}, \vec{Y}_{-l})$ denotes the sample data with $l$ lags apart, which consists of $(n - l)$ pairs of observations:

$$(\vec{X}, \vec{Y}_{-l}) = \{(X_{1+l}, Y_1), (X_{2+l}, Y_2), \dots, (X_n, Y_{n-l})\}.$$

**Step 2:** Estimate the optimal dependence lag:

$$\hat{L}^* = \arg \max_{l \in [0, L]} \left( \frac{n - l}{n} \right) \cdot \tau_n(\vec{X}, \vec{Y}_{-l}).$$

Here, the weight $\left( \frac{n-l}{n} \right)$ simply weights each cross dependence statistic based on the number of observations it uses.

**Step 3:** Compute the temporal dependence sample statistic:

$$\mathrm{T}_n(\vec{X}, \vec{Y}) = \sum_{l=0}^{L} \left( \frac{n - l}{n} \right) \cdot \tau_n(\vec{X}, \vec{Y}_{-l}).$$

**Step 4:** Compute the p-value using block permutation:

$$\text{p-val} = \sum_{r=1}^{R} \mathrm{I}(\mathrm{T}_n(\vec{X}, \vec{Y}) > \mathrm{T}_n(\vec{X}, \vec{Y}_{\pi_B}))/R,$$

where $\mathrm{I}(\cdot)$ is the 0-1 indicator function, and $\pi_B$ is a randomly generated block permutation for each $r$.

**Output:** The temporal dependence statistic T, the corresponding p-value, and the estimated optimal dependence lag $\hat{L}^*$.

The null hypothesis is rejected if the p-value is less than a pre-specified Type 1 error level, such as 0.05.

## 2.3 Choice of Dependence Measure

While the algorithm can accommodate any dependence measure as the choice of $\tau_n(\cdot, \cdot)$, it is essential for the chosen measure to be well-behaved and satisfy the required assumptions outlined in Section 3.1. This ensures consistency in detecting dependence between temporal data, both in terms of performance and subsequent theory. In our experiments, we employed distance correlation, Hilbert-Schmidt independence criterion, and multiscale graph correlation. All of these measures meet the necessary assumptions, and the resulting tests appear valid and consistent in our numerical experiments.

As the proposed temporal statistic is essentially an aggregation of the underlying dependence measure, its effectiveness in capturing dependence is contingent upon the choice of dependence measure. It is well known that each dependence measure has its own unique strengths. Therefore, our usage of distance and kernel based statistics in this paper should be viewed as an illustration of the validity and consistency properties of the proposed temporal test.

Some examples of other dependence measures include correlation coefficients (Fukumizu et al., 2007; Bießmann et al., 2010), Chatterjee's rank correlation (Chatterjee, 2021; Shi et al., 2021; 2022), the HHG method (Heller et al., 2013; 2016), projection correlation (Zhu et al., 2017), ball covariance (Pan et al., 2020), as well as recent high-dimensional dependence statistics (Zhu et al., 2020; Huang & Huo, 2022; Shen & Dong, 2024; Xu et al., 2024; Zhou et al., 2024). All of these dependence measures can be directly incorporated into our temporal testing framework by simply modifying the cross-dependence statistics in Step 1. Such adaptations may offer better testing power for certain dependence structures.

For instance, using the correlation coefficient with block permutation will only detect linear associations in temporal data, while a universally consistent dependence measure can detect all possible dependencies with a sufficiently large sample size; dependence measures that are better at detecting nonlinear or high-dimensional dependencies in standard i.i.d. data will also perform better under such dependencies in the case of temporal data, requiring a smaller sample size to achieve perfect testing power; rank-based dependence measures can be more robust against data noise.

## 2.4   The Block Permutation Test

The standard permutation test is widely used for independence testing (Good, 2005). In a standard permutation, $\pi(\cdot)$ randomly permutes the indices $1, 2, \ldots, n$, resulting in $\vec{Y}_\pi$ and $\vec{X}$ that are mostly independent (except for a few indices that do not change position, which are asymptotically negligible as $n$ increases). Given sufficiently many random permutations, this process allows the permuted test statistics to estimate the true null distribution.

However, the above is only true under the standard i.i.d. setting, and it no longer holds when there exists structural dependence within the sample sequence, such as when $(X_t, Y_t)$ are dependent with $(X_{t-1}, Y_{t-1})$. Specifically, the permuted statistics would under-estimate the true null distribution, leading to an inflation of the testing power. This issue has been noted in Guillot & Rousset (2013); DiCiccio & Romano (2017), which can affect any dependence measure that relies on the standard permutation test.

To ensure validity of the test, we employ a block permutation procedure (Politis, 2003) denoted as $\pi_B(\cdot)$, where $B$ denotes the number of blocks. The construction of $\pi_B(\cdot)$ proceeds as follows:

We partition the index list into $B$ consecutive blocks. For $j = 1, \ldots, B$, block $j$ consists of indices

$$B_j = (\lceil \frac{n}{B} \rceil * (j-1) + 1, \lceil \frac{n}{B} \rceil * (j-1) + 2, \ldots, \lceil \frac{n}{B} \rceil * j - 1).$$

Note that for the last block, the last few indices may exceed $n$, in which case the indices wrap around and restart from 1.

As an example, consider a sample size of $n = 100$ and $B = 20$ blocks, with each block containing 5 indices. Then the first block would be $(Y_1, Y_2, ..., Y_5)$, the second block would be $(Y_6, Y_7, ..., Y_{10})$, etc. During the block permutation process, each block is shifted to another position. For instance, the first block might be permuted to the fourth block, resulting in $\pi_B(1) = 16, \pi_B(2) = 17, \pi_B(3) = 18, \pi_B(1) = 19, \pi_B(1) = 20$. This shuffling of blocks ensures a randomized distribution of data while maintaining the block structure.

## 2.5   Parameter Choice and Computational Complexity

The choice of the maximum lag, denoted as $L$, is typically determined based on subject matter considerations. For example, if the signal from one region of the brain can only influence another region within a range of 20 time steps, then setting $L = 20$ would be appropriate. Similarly, when collecting daily stock trading data for two stocks, choosing $L = 30$ indicates that we are examining the dependence structure within the past month.

As for the number of blocks, we used $B = 20$ in our experiments, which is sufficient for our purposes. For the number of permutation, we used $R = 1000$ replicates. Assuming that the dependence measure can be computed in $O(n^2)$ time complexity (which is the case for distance correlation), the temporal independence test has a time complexity of $O(n^2 RL)$.

## 3 Supporting Theory

In this section, we establish the asymptotic properties of the test statistics and the resulting tests, which include asymptotic convergence, validity, and consistency. We begin by outlining the necessary assumptions for the theoretical results, followed by detailed elaborations on each assumption. All theorem proofs can be found in the Appendix Section B.

### 3.1 Assumptions

- The observed data $\{(X_t, Y_t)\}_{t=1}^n$ is strictly stationary, non-constant, and the underlying distribution $F_{XY_{-l}}$ has finite moments for any lag $l \geq 0$.

- There exists a maximum dependence lag $M$ such that for all $l \geq M$, the two time series are almost independent for large $n$, so are each time series within itself:

$$\sup |F_{XY_{-l}} - F_X F_Y| = O(\frac{1}{n}),$$
$$\sup |F_{XX_{-l}} - F_X F_X| = O(\frac{1}{n}),$$
$$\sup |F_{YY_{-l}} - F_Y F_Y| = O(\frac{1}{n}).$$

- The maximum dependence lag $M$ and the maximum lag under consideration $L$ are non-negative integers that satisfies $L \geq M$ and $L = o(n)$, i.e., they may increase together with $n$ but at a slower pace.

- As the sample size $n$ increases, both the number of blocks $B$ and the number of observations per block $\frac{n}{B}$ increase to infinity. Moreover, $\frac{n}{B} \geq M$ for sufficiently large $n$.

- The sample dependence measure has the following form:

$$\tau_n(\vec{X}, \vec{Y}) = \frac{\sum_{i=1}^n \sum_{j=1}^n \gamma_n(i, j)}{n^2},$$

  where each $\gamma_n(i, j)$ is a function of $(X_i, X_j, Y_i, Y_j)$, and remaining sample pairs may also be used but with a weight of $O(1/n)$.

- In the standard i.i.d. setting where $(X_1, Y_1), (X_2, Y_2), \ldots, (X_n, Y_n) \overset{i.i.d.}{\sim} F_{XY}$, there exists a population statistic $\tau(X, Y)$ defined solely based on the joint distribution $F_{XY}$. When $i \neq j$, each term in the sample statistic satisfies:

$$\mathbb{E}(\gamma_n(i, j)) = \tau(X, Y) + o(1).$$

  Moreover, the population statistic $\tau(X, Y)$ is non-negative and equals 0 if and only if $X$ and $Y$ are independent, i.e., $F_{XY} = F_X F_Y$.

The first assumption is a common one in time series research. The key distinction from the standard i.i.d. setting is that the samples are no longer independent, but remain identically distributed. For non-stationary data, there exist many common techniques to remove trends and process them into approximately stationary processes (Cleveland et al., 1990; Hastie et al., 2009; Enders, 2010; Shumway & Stoffer, 2010; Box et al., 2015). Some examples include differencing, where one computes the difference between consecutive

observations; detrending via linear regression or polynomial fitting and subtracting the trend component from the original series; seasonal adjustment by decomposition; log / square root / Box-Cox transformation to stabilize variance; smoothing via moving averages to reduce noise and short-term fluctuations; filtering to remove specific frequencies from the data.

The second and third assumptions require that the time series exhibit independence for sufficiently large lags beyond $M$, and that the maximum lag to be examined, $L$, must be no less than $M$. Such an assumption shares similarity with the mixing property, where a stochastic process is mixing if its values at widely-separated times are asymptotically independent (Pham & Tran, 1985; McDonald et al., 2011; Ziemann & Tu, 2022). Hence, our results can also be considered approximately true for mixing time series.

The fourth assumption imposes a regularity condition on block permutation. In theory, choices for $B$ can be $log(n)$ or $\sqrt{n}$, while a practical choice like $B = 20$ is sufficient for our simulations. This resembles the Bayes optimal condition for K-nearest-neighbor, where $K$ is required to increase to infinity but slower than $n$.

The remaining assumptions regarding the dependence measure are satisfied by a variety of distance and kernel measures that have been recently proposed. For example, distance covariance satisfies the two assumptions, with

$$\gamma_n(i,j) = \{d(X_i, X_j) - \mu_{X_i} - \mu_{X_j} + \mu_X\}\{d(Y_i, Y_j) - \mu_{Y_i} - \mu_{Y_j} + \mu_Y\}.$$

Here, $d(\cdot, \cdot)$ is the Euclidean distance, $\mu_{X_i}$ denotes the mean of all distance pairs relative to $X_i$ within $\vec{X}$, and $\mu_X$ is the mean of the whole pairwise distance matrices of $\vec{X}$. Furthermore, the population distance covariance is defined in terms of characteristic functions and equals 0 if and only if $F_{XY} = F_X F_Y$ in the standard i.i.d. settings. Indeed, many dependence measures that are universal consistent in the standard i.i.d. setting satisfy this assumption. For example, the Hilbert-Schmidt independence criterion utilizes the same formulation (Shen & Vogelstein, 2021; Sejdinovic et al., 2013) on the Gaussian kernel. Additionally, the unbiased distance covariance and distance correlation, as well as the multiscale graph correlation – a truncated version of distance correlation where large distance pairs may be unused – also satisfy this assumption.

### 3.2 Convergence of the Sample Statistics

We begin by proving the convergence of the sample cross dependence to the population cross dependence:

**Theorem 1.** *The cross dependence sample statistic satisfies:*

$$\mathbb{E}(\tau_n(\vec{X}, \vec{Y}_{-l})) - \tau(X, Y_{-l}) = o(1),$$

$$Var(\tau_n(\vec{X}, \vec{Y}_{-l}))) = O(\frac{1}{n-l}).$$

*Therefore, for each $l \in \{0, ..., L\}$, we have*

$$\tau_n(\vec{X}, \vec{Y}_{-l}) \overset{n \to \infty}{\Rightarrow} \tau(X, Y_{-l})$$

*in probability.*

Theorem 1 shows that both the bias and variance of the cross dependence statistic diminish to 0 as the sample size $n$ increases. Consequently, this guarantees that the aggregated temporal dependence statistic and the estimated optimal lag also converge to their corresponding population forms in probability.

**Theorem 2.** *The temporal dependence sample statistic satisfies:*

$$\mathrm{T}_n(\vec{X}, \vec{Y}) \overset{n \to \infty}{\Rightarrow} \sum_{l=0}^{L} \tau(X, Y_{-l}).$$

*The estimated optimal dependence lag satisfies:*

$$\hat{L}^* \overset{n \to \infty}{\Rightarrow} \arg \max_{l \in [0, L]} \tau(X, Y_{-l}).$$

### 3.3 Validity and Consistency for Testing Temporal Independence

In this subsection we establish the validity and consistency of the method. Specifically, if $\vec{X}$ and $\vec{Y}$ are independent, the power of the test equals the Type 1 error level $\alpha$. Conversely, if $\vec{X}$ and $\vec{Y}$ are dependent, the power of the test converges to 1, and the method can consistently detect any dependence.

Given $\mathrm{T}_n(\vec{X}, \vec{Y})$ as the observed test statistic, let $F_{T_n^B}(z)$ be the empirical distribution of the block-permuted statistics $\{\mathrm{T}_n(\vec{X}, \vec{Y}_{\pi_B})\}$, and denote $z_{n,\alpha}$ as the critical value where:

$$F_{T_n^B}(z)(z_{n,\alpha}) = 1 - \alpha.$$

The following theorem establishes the asymptotic validity of our block permutation test:

**Theorem 3** (Asymptotic Validity)**.** *Under the null hypothesis that $\vec{X}$ and $\vec{Y}$ are independent for all lags $l \in [0, L]$, the test statistic satisfies:*

$$\mathrm{T}_n(\vec{X}, \vec{Y}) \overset{n \to \infty}{\Rightarrow} 0.$$

*Moreover, the block-permutation test is asymptotically valid, i.e.,*

$$Prob(\mathrm{T}_n(\vec{X}, \vec{Y}) \geq z_{n,\alpha}) \overset{n \to \infty}{\Rightarrow} \alpha.$$

The next theorem proves that the method is universally consistent against any alternative.

**Theorem 4** (Testing Consistency)**.** *Under the alternative hypothesis that $\vec{X}$ and $\vec{Y}$ are dependent for some lag $l \in [0, L]$, the test statistic satisfies*

$$\mathrm{T}_n(\vec{X}, \vec{Y}) \overset{n \to \infty}{\Rightarrow} c > 0.$$

*Moreover, the block-permutation test is asymptotically consistent, i.e.,*

$$Prob(\mathrm{T}_n(\vec{X}, \vec{Y}) \geq z_{n,\alpha}) \overset{n \to \infty}{\Rightarrow} 1.$$

## 4 Simulations

We estimated the testing power of the proposed approach through simulations on various temporal dependence structures. Specifically, we considered three different implementations of the proposed temporal dependence statistic, which utilized distance correlation (`DCorr`), Hilbert-Schmidt independence criterion (`HSIC`), and multiscale graph correlation (`MGC`). For comparison, we included `ShiftHSIC` (Chwialkowski & Gretton, 2014), `WildHSIC` (Chwialkowski et al., 2014), and the widely recognized `Ljung-Box` test (Ljung & Box, 1978) using traditional cross-correlations. Each simulation was repeated 300 times, with 1000 permutations and a Type 1 error level of $\alpha = 0.05$ used to compute the $p$-values. The testing power is measured by how often the p-value is lower than 0.05 out of the 300 Monte-Carlo simulations. Analysis of `ShiftHSIC` and `WildHSIC` was performed using MATLAB code[1] and `wildBootstrap`[2].

### 4.1 Testing Power Evaluation

**Independence** First, we check the validity of the tests by generating two independent, stationary autoregressive time series with a lag of one:

$$\begin{bmatrix} X_t \\ Y_t \end{bmatrix} = \begin{bmatrix} \phi & 0 \\ 0 & \phi \end{bmatrix} \begin{bmatrix} X_{t-1} \\ Y_{t-1} \end{bmatrix} + \begin{bmatrix} \epsilon_t \\ \eta_t \end{bmatrix}.$$

Here, $(\epsilon_t, \eta_t)$ are standard normal noise terms. As shown in Figure 1, the proposed methods maintain a testing power close to $\alpha = 0.05$ across varying $n$ and $\phi$, regardless of the statistic used.

---

[1] https://github.com/kacperChwialkowski/HSIC/
[2] https://github.com/kacperChwialkowski/wildBootstrap

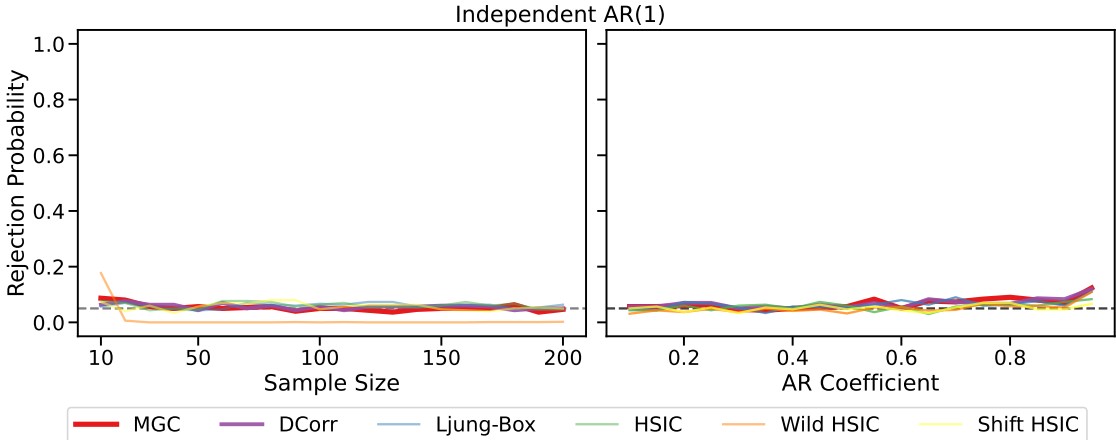

Figure 1: This figure illustrates the validity of the tests using two independent time series. In the left panel, the testing power is computed as the sample size increases, with an AR coefficient of $\phi = 0.5$. The right panel keeps the sample size at $n = 1200$ while varying the AR coefficient $\phi$, with the noise variance appropriately adjusted by $(1 - \phi^2)$, based on the same simulation as in Chwialkowski & Gretton (2014). The dashed black line represents the significance level $\alpha = 0.05$.

**Linear Dependence**   Next, we assess our methods' ability to capture linear relationships in the following simulation:

$$\begin{bmatrix} X_t \\ Y_t \end{bmatrix} = \begin{bmatrix} 0 & \phi \\ \phi & 0 \end{bmatrix} \begin{bmatrix} X_{t-1} \\ Y_{t-1} \end{bmatrix} + \begin{bmatrix} \epsilon_t \\ \eta_t \end{bmatrix}.$$

As this represents a straightforward linear relationship, the `Ljung-Box` test, based on auto-correlation, is expected to perform best. This is indeed the case in the left panel of Figure 2. Our proposed methods using `DCorr`, `MGC`, and `HSIC` follow closely, quickly converging to perfect power around $n = 100$. In contrast, the other competitors do not perform well in this scenario. This is not surprising, as the `ShiftHSIC` method is designed to detect whether $X_t$ and $Y_t$ are dependent at lag 0, whereas the linear dependence here is of lag 1. The `WildHSIC` method used a wild bootstrap method to estimate the null distribution, which can be inaccurate at small sample size.

**Nonlinear Dependence**   The next simulation considers a nonlinear dependent model:

$$\begin{bmatrix} X_t \\ Y_t \end{bmatrix} = \begin{bmatrix} \epsilon_t Y_{t-1} \\ \eta_t \end{bmatrix}.$$

In the right panel of Figure 2, our proposed methods utilizing `DCorr`, `MGC`, and `HSIC` demonstrate superior performance compared to other competing methods. Notably, the `HSIC` and `MGC` implementations exhibit better finite-sample power, as these two dependence measures are better at identifying nonlinear relationships than `DCorr`. In contrast, all other tests fail to detect dependence in this scenario.

**Extinct Gaussian**   This simulation uses the same extinct Gaussian process from Chwialkowski & Gretton (2014), where

$$\begin{bmatrix} X_t \\ Y_t \end{bmatrix} = \begin{bmatrix} \phi & 0 \\ 0 & \phi \end{bmatrix} \begin{bmatrix} X_{t-1} \\ Y_{t-1} \end{bmatrix} + \begin{bmatrix} \epsilon_t \\ \eta_t \end{bmatrix},$$

and we set $n = 1200$. Here, the $(\epsilon_t, \eta_t)$ pair are dependent and drawn from an Extinct Gaussian distribution with two additional parameters: $e$ (extinction rate) and $r$ (radius). Both variables are initially drawn from independent standard normal, and $U$ is sampled from standard uniform. If either $\epsilon_t^2 + \eta_t^2 > r$ or $U > e$ holds, then $(\epsilon_t, \eta_t)$ are returned; otherwise, they are discarded and the process is repeated. In this process, the dependence between $\epsilon_t$ and $\eta_t$ increases with extinction rate $e$. Therefore, we expect power to increase with the extinction rate, which is indeed the case as shown in Figure 3. While all methods, except `Ljung-Box`,

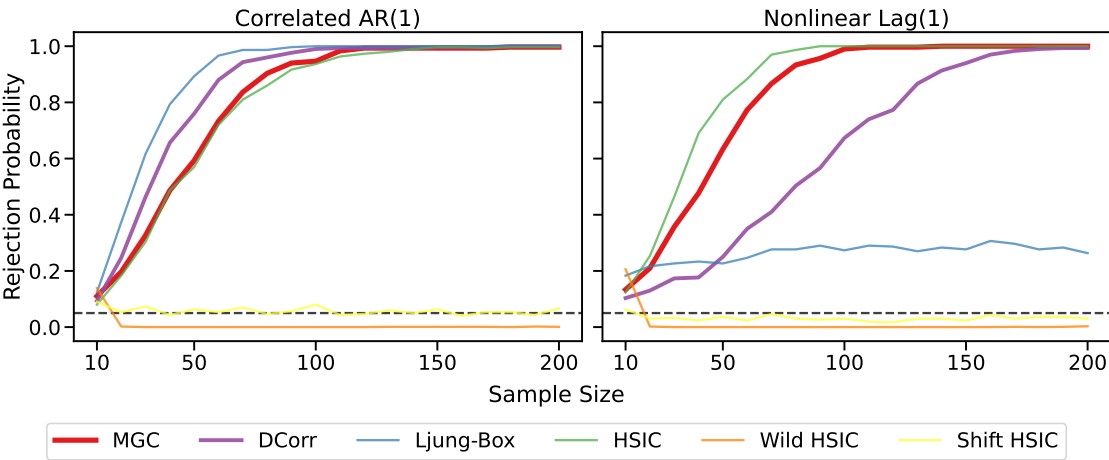

Figure 2: The testing power for linear (left panel) and nonlinear (right panel) simulations based on 300 replicates.

are consistent and eventually achieve perfect power, our proposed method using `MGC` stands out as the best performer.

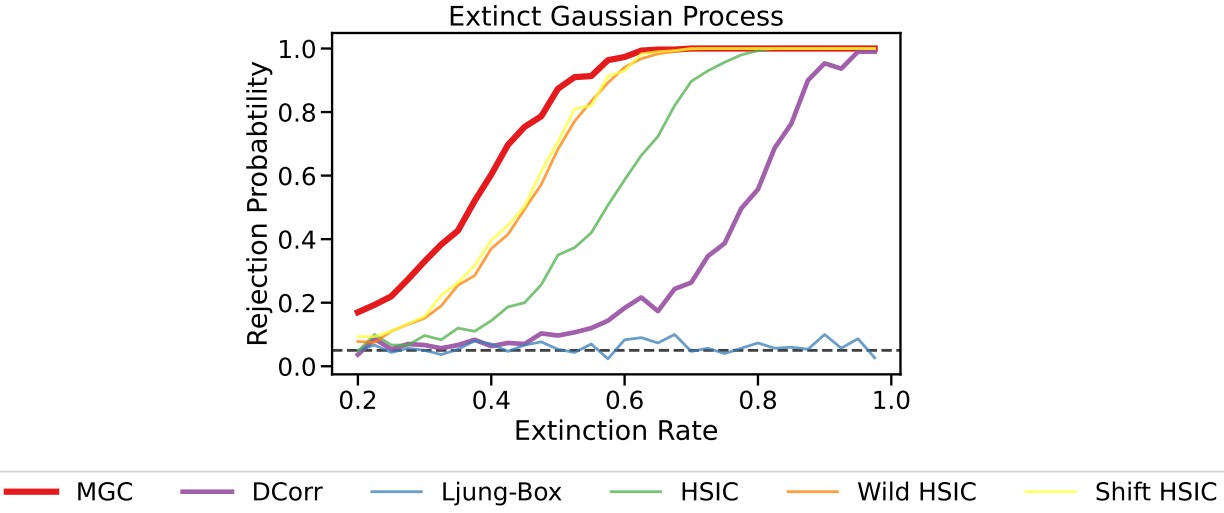

Figure 3: The testing power for the extinct gaussian simulation based on 300 replicates.

## 4.2   Optimal Dependence Lag Estimation

In this subsection, we evaluate the method's performance in estimating the optimal dependence lag in both linear and nonlinear settings. The linear setting is

$$\begin{bmatrix} X_t \\ Y_t \end{bmatrix} = \begin{bmatrix} 0 & \phi_1 \\ \phi_1 & 0 \end{bmatrix} \begin{bmatrix} X_{t-1} \\ Y_{t-1} \end{bmatrix} + \begin{bmatrix} 0 & \phi_3 \\ \phi_3 & 0 \end{bmatrix} \begin{bmatrix} X_{t-3} \\ Y_{t-3} \end{bmatrix} + \begin{bmatrix} \epsilon_t \\ \eta_t \end{bmatrix},$$

where we set $\phi_3 = 0.8 > \phi_1 = 0.1$ such that the true optimal dependence lag equals 3. The nonlinear simulation is

$$\begin{bmatrix} X_t \\ Y_t \end{bmatrix} = \begin{bmatrix} \epsilon_t Y_{t-3} \\ \eta_t \end{bmatrix}.$$

In both simulations, the true optimal dependence lag equals 3. Figure 4 shows that the proposed method using either `DCorr` or `MGC` consistently estimates the optimal dependence lag as the sample size increases, and `MGC` outperforms `DCorr` in the nonlinear setting.

### 4.3 Multivariate Simulations

In this subsection, we revisit the testing power and dependence lag estimation in both linear and nonlinear settings, maintaining a fixed sample size of $n = 100$ and increasing the dimensionality $p$, to evaluate performance for multivariate data.

For testing power evaluation, we use the following multivariate linear setting:

$$\begin{bmatrix} X_t \\ Y_t \end{bmatrix} = \begin{bmatrix} 0 & \phi D \\ \phi D & 0 \end{bmatrix} \begin{bmatrix} X_{t-1} \\ Y_{t-1} \end{bmatrix} + \begin{bmatrix} \epsilon_t \\ \eta_t \end{bmatrix},$$

where $\phi = 0.65$, $D \in \mathbb{R}^{p \times p}$ is a diagonal matrix where the elements are $D_{ii} = 1/i$, and $\epsilon_t, \eta_t$ are standard normal of dimension $p$. In a similar manner, we use the following multivariate nonlinear setting:

$$\begin{bmatrix} X_t \\ Y_t \end{bmatrix} = \begin{bmatrix} D(\epsilon_t \odot Y_{t-1}) \\ \eta_t \end{bmatrix},$$

where $\odot$ denotes element-wise multiplication. We intentionally design the matrix $D$ as a decaying weight, reflecting a meaningful multivariate simulation where additional dimensions contain weaker dependence signals.

Figure 5 illustrates the testing power as dimensionality increases. At a fixed sample size, all testing powers gradually decrease as $p$ increases. The proposed method using any of `MGC`, `DCorr`, or `HSIC` maintains relatively stable power with slow degradation in the case of linear dependence. The same trend is observed for nonlinear dependence, although the degradation is faster, with the `MGC` statistic performing the best. It is worth emphasizing that due to the consistent property, if we fix $p$ and let $n$ increase, the testing power for our method shall increase to 1.

Similarly, we extend the optimal lag estimation into the following two multivariate settings:

$$\begin{bmatrix} X_t \\ Y_t \end{bmatrix} = \begin{bmatrix} 0 & \phi_1 D \\ \phi_1 D & 0 \end{bmatrix} \begin{bmatrix} X_{t-1} \\ Y_{t-1} \end{bmatrix} + \begin{bmatrix} 0 & \phi_3 D \\ \phi_3 D & 0 \end{bmatrix} \begin{bmatrix} X_{t-3} \\ Y_{t-3} \end{bmatrix} + \begin{bmatrix} \epsilon_t \\ \eta_t \end{bmatrix}.$$
$$\begin{bmatrix} X_t \\ Y_t \end{bmatrix} = \begin{bmatrix} D(\epsilon_t \odot Y_{t-3}) \\ \eta_t \end{bmatrix},$$

where $\phi_1 = 0.1$ and $\phi_3 = 0.65$. These settings are similar to those in Section 4.2, with the addition of increasing dimension and $D$. The true optimal lag remains 3. The estimation accuracy, as shown in Figure 6, demonstrates successful detection at small $p$, with accuracy gradually degrading as $p$ increases.

## 5 Real Data

### 5.1 Analyzing Connectivity in the Human Brain

This study is based on data from an individual (Subject ID: 100307) of the Human Connectome Project (HCP), which can be downloaded online[3]. The human cortex is parcellated into 180 parcels per hemisphere using the HCP multi-modal parcellation atlas (Glasser et al., 2016). For this study, 22 parcels were selected as regions of interest (ROIs), representing various locations across the cortex. These parcels are denoted as $X^{(1)}, \ldots, X^{(22)}$. Each parcel consists of a contiguous set of vertices whose fMRI signal is projected on the cortical surface. Averaging the vertices within a parcel yields a univariate time series $X^{(u)} = (X_1^{(u)}, \ldots, X_n^{(u)})$, where $n = 1200$ in this particular case. The selected ROIs, their parcel number in the HCP multi-modal parcellation (Glasser et al., 2016), and assigned network are listed in Table 1.

---

[3]https://www.humanconnectome.org/study/hcp-young-adult/data-releases

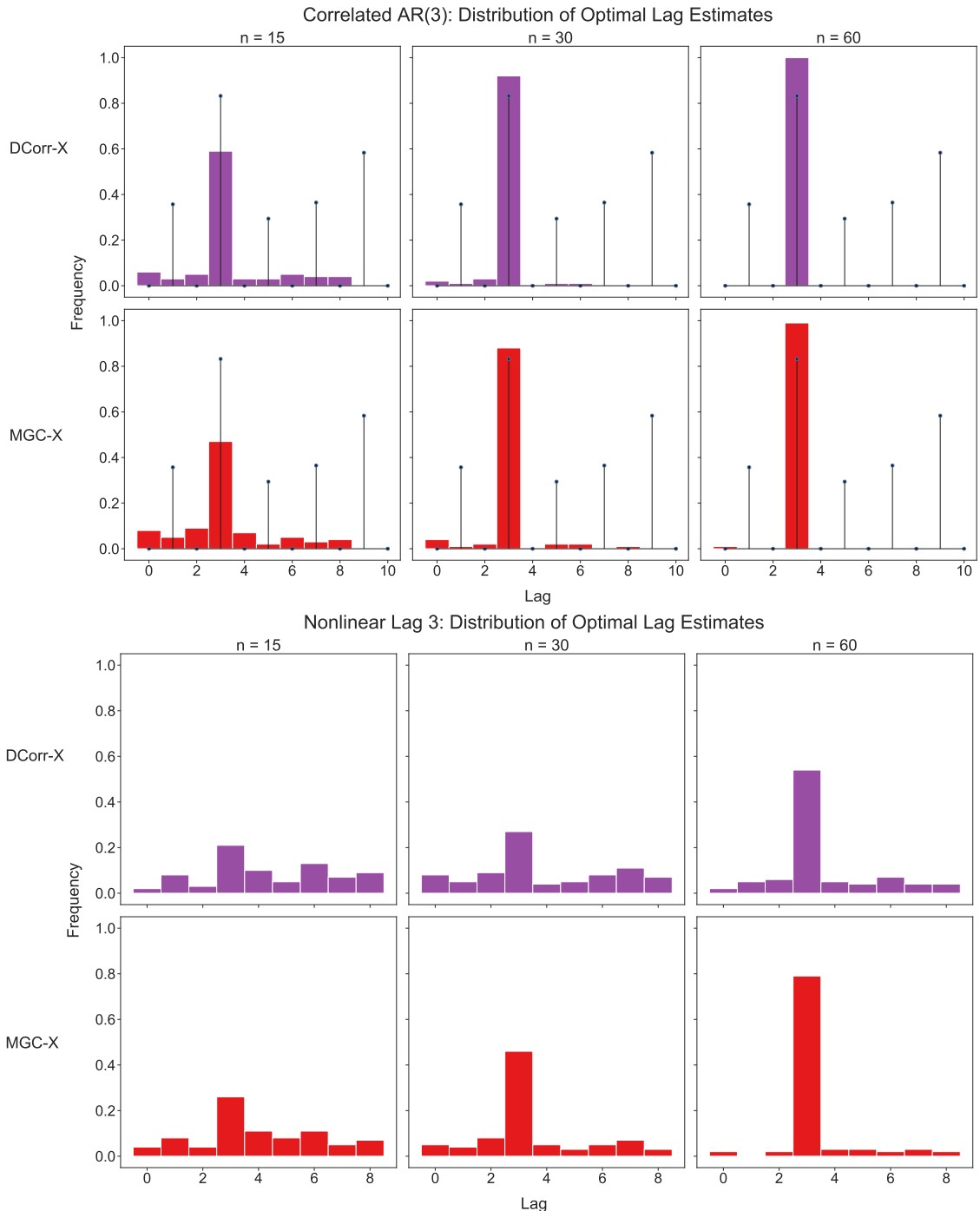

Figure 4: This figure displays the performance of our proposed method using both `MGC` and `DCorr` for estimating the optimal dependence lag $\hat{L}^*$ in linear and nonlinear relationships. The colored bar above lag $l$ shows the empirical frequency of $\hat{L}^* = j$, with red representing `MGC` and purple representing `DCorr`. The probability is estimated based on 100 trials. The first row shows `DCorr` estimation performance at sample sizes $n = 15, 30, 60$ for linear relationships, while the second row shows the `MGC` performance on the same data. The third row displays `DCorr` estimation for nonlinear relationships, and the last row presents the same for `MGC`.

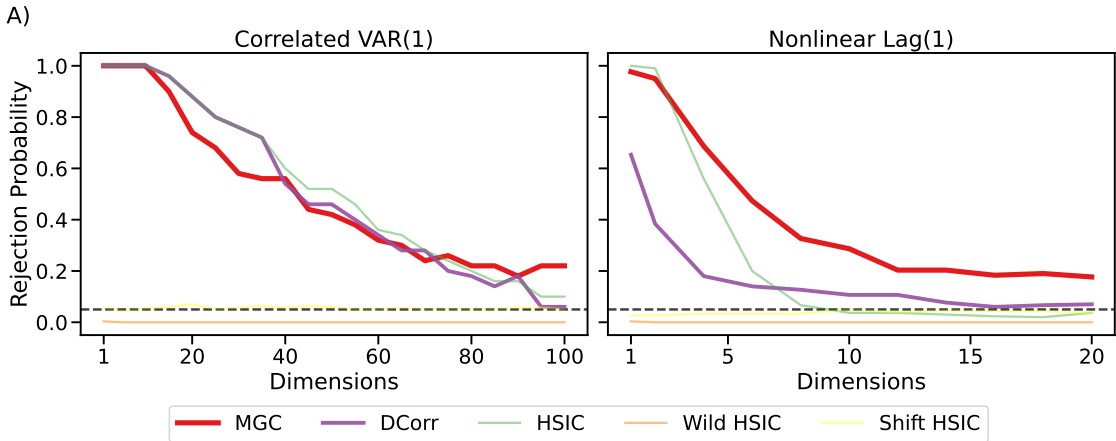

Figure 5: This figure shows the testing power for multivariate simulations, with a constant sample size of $n = 100$ while increasing the dimensionality.

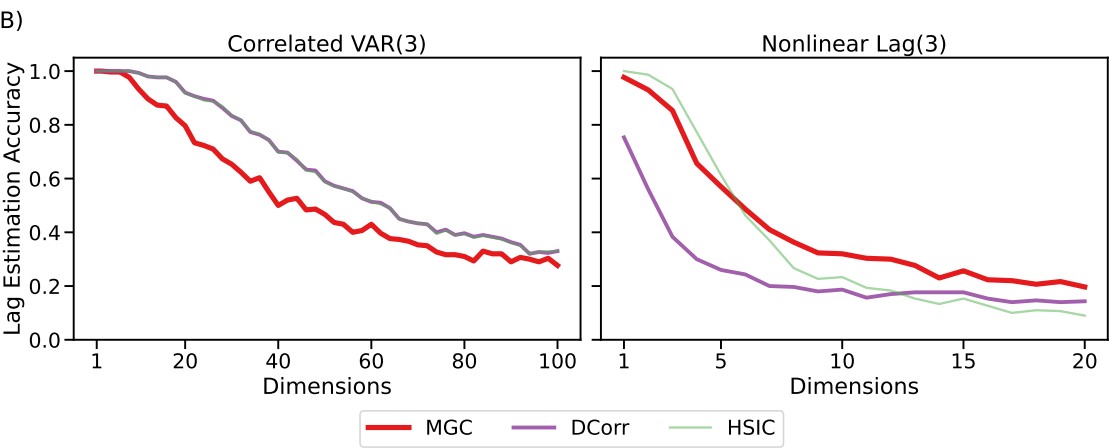

Figure 6: This figure shows the accuracy of estimating the true optimal lag in multivariate simulations, maintaining a constant sample size of $n = 100$ while increasing the dimensionality. Note that in the left panel, the lines representing DCorr and HSIC overlap due to their almost identical performance.

| ROI ID | Network | Shorthand | Parcel Key | Parcel Name |
|--------|---------|-----------|------------|-------------|
| 18 | Default Mode Network | DMN | 150 | PGi |
| 19 | Default Mode Network | DMN | 65 | p32pr |
| 20 | Default Mode Network | DMN | 161 | 32pd |
| 21 | Default Mode Network | DMN | 132 | TE1a |
| 22 | Default Mode Network | DMN | 71 | 9p |
| 6 | Dorsal Attention Network | dAtt | 96 | 6a |
| 7 | Dorsal Attention Network | dAtt | 117 | API |
| 8 | Dorsal Attention Network | dAtt | 50 | MIP |
| 9 | Dorsal Attention Network | dAtt | 143 | PGp |
| 10 | Ventral Attention Network | vAtt | 109 | MI |
| 11 | Ventral Attention Network | vAtt | 148 | PF |
| 12 | Ventral Attention Network | vAtt | 60 | p32pr |
| 13 | Ventral Attention Network | vAtt | 38 | 23c |
| 1 | Visual Network | Visual | 1 | V1 |
| 2 | Visual Network | Visual | 23 | MT |
| 3 | Visual Network | Visual | 18 | FFC |
| 16 | FrontoParietal Network | FP | 83 | p9-46v |
| 17 | FrontoParietal Network | FP | 149 | PFm |
| 14 | Limbic Network | Limbic | 135 | TF |
| 15 | Limbic Network | Limbic | 93 | OFC |
| 4 | Somatomotor Network | SM | 53 | 3a |
| 5 | Somatomotor Network | SM | 24 | A1 |

Table 1: This table displays the parcellation information for the parcels used in our analysis. They are listed based on the numeric order they appear in Figure 7.

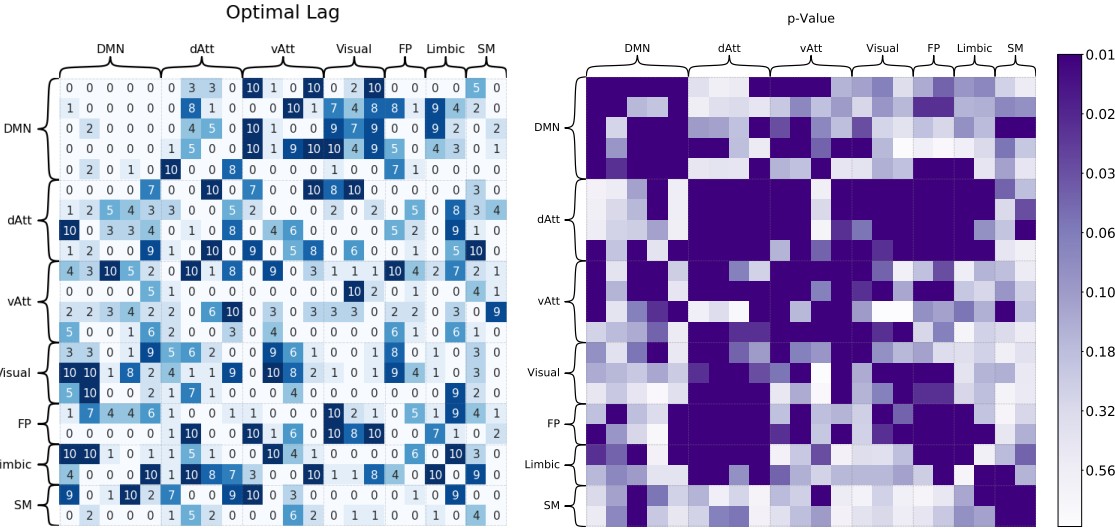

Figure 7: This displays the results of applying the temporal dependence method using `MGC` to resting-state fMRI data.

As the temporal dependence method using `MGC` performed well in our simulations, we simply use the `MGC` implementation in this analysis. In the left panel of Figure 7, we present the optimal dependence lag for each interdependency, ranging up to $L = 10$. Meanwhile, the right panel of Figure 7 displays the log-scale $p$-values of temporal dependence for each pair of parcels. Generally, we observe strong relationships with small lags within the same region, such as an optimal lag of usually 0 within the "DMN" region with significant p-values. In contrast, inter-region dependencies are less significant and typically exist at longer lags.

## 5.2 Discovering Temporal Dependence Structure of Low-Beta Stocks

In this experiment, we apply the proposed methodology to analyze the financial market and uncover interesting nonlinear dependencies between low-beta stocks and the S&P 500. In the US financial market, it is well-known that almost all stocks are linearly related to the broad market. A commonly used statistic to measure this association is the beta value, which quantifies a stock's volatility relative to the market (S&P 500). Beta is defined as the covariance between an individual stock and the general market, divided by the variance of the general market, and it utilizes the rate of return per month rather than the stock price. A beta less than 1 suggests that the stock is less volatile than the market, while a beta greater than 1 indicates higher volatility.

Low-beta stocks are an interesting concept in investing, defined by having a relatively small beta value, typically around 0.5 or less. These stocks tend to have lower correlations with market movements compared to high-beta stocks and are often associated with companies that have stable earnings, strong cash flows, and less uncertainty about their future prospects. Therefore, low-beta stocks can play a valuable role in a well-diversified investment portfolio by providing stability, reducing risk, and potentially enhancing long-term performance. Moreover, with the right strategy, stocks with low volatility can generate high risk-adjusted returns (Blitz & Vliet, 2007; Frazzini & Pedersen, 2014).

We collected weekly closing stock prices from January 1, 2014, to May 1, 2014, using Yahoo Finance data, for the S&P 500 ETF (the benchmark) and 10 individual stocks, as shown in Figure 8. In addition to NVDA, AAPL, and MSFT, which are mega cap stocks and were included for comparison purposes, the remaining stocks are commonly found in low-beta portfolios. Given the high volatility of daily stock prices, we chose to collect the closing price per week, and process each stock's weekly price into rates of return to make the data resemble a stationary sequence.

As the stock market is highly related and linear relationships are dominant among stocks, we chose to use Pearson correlation and distance correlation as our choice of dependence measures. For each choice, we computed the cross-dependence measures between each individual stock and the S&P 500 from lag 0 to 4. Subsequently, we computed the aggregated temporal statistic, followed by optimal lag estimation and p-value computation using block permutations. The sample size is $n = 538$, and the number of blocks is 20. Therefore, our aim is to test the existence of temporal dependence between each individual stock and the general market, from concurrent testing to a lag of up to 1 month.

For each individual stock, the aggregated test always yielded a significant result, with a p-value of 0 and an optimal lag of 0 in every case. This indicates that all individual stocks are dependent on the general market, with the strongest dependence observed at concurrent (lag 0) intervals. This outcome is not surprising, as even a beta of 0.3 readily implies a significant linear relationship at lag 0.

Next, we examine the cross-lag dependence structure and their individual p-values in block-permutation, as reported in Figure 9. We first consider the Pearson correlation: the lag 0 concurrent testing statistic is very similar to the beta value of each stock, all of which have significant p-values of 0. On the other hand, all other lags have insignificant p-values, suggesting there is no linear relationship beyond lag 0. Namely, the S&P 500 price this week has no linear effect on itself next week or any individual stock next week, e.g., a high return of $+2\%$ this week does not always imply another $+2\%$ next week.

Inspecting the distance correlation measure yields interesting new insights: the general market, the three mega-cap stocks, and a few low-beta stocks are actually dependent on the general market at lag 1 and beyond. Since the Pearson correlation indicates the lack of a linear relationship, this dependence must be nonlinear, which is weakened as the lag increases. This insight aligns with empirical experience: for example, a high return of $+2\%$ this week may indicate that the next week will be volatile, potentially resulting in another week of high return or a significant pullback from the highs. Such dependence constitutes a nonlinear association, while the linear association could be almost 0. This type of volatility is often utilized in option trading and holds promise for future applications.

Finally, Figure 9 also reveal that some low-beta stocks, such as T, JNJ, WMT, and LLY, exhibit independence from the general market beyond the concurrent lag 0. This insight suggests that these stocks could be ideal

| Company | Stock Symbol | Beta (5 Year Monthly) |
|---|---|---|
| SP500 ETF | SPY | 1 |
| Nvidia | NVDA | 1.75 |
| Apple Inc | AAPL | 1.26 |
| Microsoft | MSFT | 0.89 |
| AT&T Inc | T | 0.7 |
| Pfizer Inc | PFE | 0.58 |
| Coca Cola | KO | 0.57 |
| Johnson & Johnson | JNJ | 0.55 |
| Walmart Inc | WMT | 0.49 |
| Procter & Gamble | PG | 0.42 |
| Eli Lily | LLY | 0.37 |

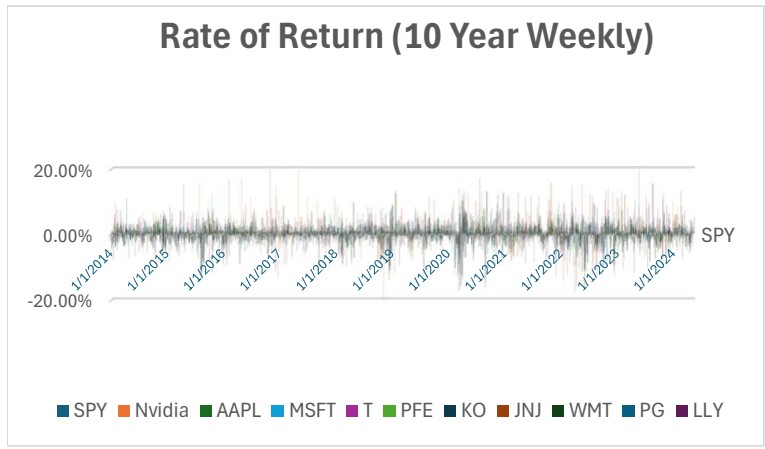

Figure 8: This figure shows the selected stocks, their current beta, and their rate of return over 10 years on a weekly basis.

candidates for a portfolio that offers temporal independence, rather than just a lack of linear relationship, from the general market.

## 6  Conclusion

This paper introduces a new independence testing procedure for temporal data. The method combined the strengths of nonparametric dependence measures, the specialized cross-lag statistic for time series, and the block permutation procedure. As a result, it provides an asymptotically valid and universally consistent approach with outstanding numerical performance. While the exposition of this manuscript is focused on time series data, this work marks an important step in extending independence testing to structural data beyond the realm of standard i.i.d. data, making them more attractive and broadly applicable.

There are several avenues for future research that warrant exploration. Firstly, although we have demonstrated the asymptotic validity of the block permutation test, its computational efficiency remains a challenge when dealing with large sample sizes. Recent studies (Zhang et al., 2018; Shen et al., 2022) have investigated faster testing procedures by approximating the null distribution of distance and kernel correlations under the standard i.i.d. setting. Extending such approaches to structural data could significantly enhance computational scalability.

| | Cross Pearson Correlation with SP500 | | | | | P-Value by Block Permutation | | | | |
|---|---|---|---|---|---|---|---|---|---|---|
| | Lag 0 | Lag 1 | Lag 2 | Lag 3 | Lag 4 | Lag 0 | Lag 1 | Lag 2 | Lag 3 | Lag 4 |
| SP500 ETF | 1.0000 | 0.0117 | -0.0335 | -0.0560 | 0.0454 | 0 | 0.782 | 0.494 | 0.172 | 0.288 |
| Nvidia | 0.5937 | 0.0148 | 0.0055 | -0.0185 | 0.0347 | 0 | 0.73 | 0.914 | 0.594 | 0.382 |
| Apple Inc | 0.6691 | -0.0195 | -0.0079 | -0.0081 | 0.0686 | 0 | 0.658 | 0.858 | 0.832 | 0.124 |
| Microsoft | 0.7177 | -0.0247 | -0.0351 | -0.0429 | 0.0541 | 0 | 0.584 | 0.44 | 0.274 | 0.202 |
| AT&T Inc | 0.3787 | -0.0677 | -0.0355 | -0.0270 | -0.0155 | 0 | 0.13 | 0.388 | 0.52 | 0.696 |
| Pfizer Inc | 0.4482 | 0.0706 | -0.0337 | -0.0087 | 0.0383 | 0 | 0.13 | 0.448 | 0.812 | 0.44 |
| Coca Cola | 0.5024 | -0.0057 | -0.0655 | -0.0755 | 0.0032 | 0 | 0.87 | 0.124 | 0.074 | 0.954 |
| Johnson & Johnson | 0.4810 | 0.0132 | -0.0536 | -0.0116 | 0.0381 | 0 | 0.782 | 0.168 | 0.766 | 0.396 |
| Walmart Inc | 0.3845 | -0.0606 | -0.0242 | -0.0165 | 0.0742 | 0 | 0.144 | 0.612 | 0.666 | 0.09 |
| Procter & Gamble | 0.4163 | -0.0196 | -0.0543 | 0.0264 | 0.0326 | 0 | 0.662 | 0.202 | 0.564 | 0.488 |
| Eli Lily | 0.2877 | 0.0032 | -0.0395 | 0.0030 | 0.0914 | 0 | 0.946 | 0.402 | 0.932 | 0.036 |
| | Cross Distance Correlation with SP500 | | | | | P-Value by Block Permutation | | | | |
| | Lag 0 | Lag 1 | Lag 2 | Lag 3 | Lag 4 | Lag 0 | Lag 1 | Lag 2 | Lag 3 | Lag 4 |
| SP500 ETF | 1.0000 | 0.0175 | 0.0114 | 0.0102 | 0.0061 | 0 | 0 | 0.008 | 0.004 | 0.022 |
| Nvidia | 0.3264 | 0.0093 | 0.0046 | 0.0059 | 0.0005 | 0 | 0.014 | 0.052 | 0.042 | 0.364 |
| Apple Inc | 0.3755 | 0.0115 | 0.0018 | 0.0047 | 0.0016 | 0 | 0.006 | 0.172 | 0.064 | 0.186 |
| Microsoft | 0.4556 | 0.0068 | 0.0013 | 0.0020 | 0.0020 | 0 | 0.03 | 0.288 | 0.14 | 0.146 |
| AT&T Inc | 0.1092 | 0.0022 | -0.0009 | -0.0018 | 0.0010 | 0 | 0.19 | 0.56 | 0.746 | 0.302 |
| Pfizer Inc | 0.1419 | 0.0071 | 0.0054 | -0.0009 | -0.0007 | 0 | 0.03 | 0.032 | 0.572 | 0.554 |
| Coca Cola | 0.1409 | 0.0056 | 0.0033 | -0.0019 | -0.0021 | 0 | 0.03 | 0.094 | 0.786 | 0.816 |
| Johnson & Johnson | 0.1608 | 0.0030 | 0.0036 | -0.0012 | -0.0005 | 0 | 0.152 | 0.094 | 0.588 | 0.498 |
| Walmart Inc | 0.1123 | 0.0028 | -0.0010 | -0.0004 | 0.0035 | 0 | 0.12 | 0.572 | 0.478 | 0.112 |
| Procter & Gamble | 0.1008 | 0.0056 | 0.0052 | -0.0005 | -0.0029 | 0 | 0.036 | 0.024 | 0.518 | 0.918 |
| Eli Lily | 0.0639 | 0.0026 | 0.0052 | 0.0036 | 0.0028 | 0 | 0.15 | 0.05 | 0.112 | 0.114 |

Figure 9: This figure shows the cross-correlation between each individual stock and the S&P 500, using Pearson correlation at the top and distance correlation at the bottom, for lags from 0 to 4, along with the corresponding p-values. Significant correlations, where larger values are better, are marked in green, while significant p-values, where smaller values are better, are marked in red.

Secondly, dependence measures are commonly employed in dimension reduction techniques, such as screening (Fan & Lv, 2008; Li et al., 2012), especially in high-dimensional data settings. However, little attention has been given to the temporal domain. While it is straightforward to utilize dependence measures for dimension reduction in multivariate time series, delving into their theoretical properties and their relationships with other standard tools, such as independence component analysis, could provide valuable insights.

Thirdly, causal inference in time series data is an important task (Haufe et al., 2010; Winkler et al., 2016). While it is widely recognized that correlation does not imply causality, recent research has demonstrated the utility of dependence and conditional dependence tests in causal inference (Cai et al., 2022; Laumann et al., 2023). Therefore, extending this framework to encompass conditional independence and causal inference may significantly advance the understanding of causal inference in time series data.

### Acknowledgment

This work was supported by the National Institutes of Health award RF1MH128696 and RO1MH120482, the National Science Foundation award DMS-1921310 and DMS-2113099, and the Defense Advanced Research Projects Agency (DARPA) Lifelong Learning Machines program through contract FA8650-18-2-7834. The authors would like to thank Sambit Panda, Hayden Helm, Benjamin Pedigo, and Bijan Varjavand for their help and discussions in preparation of the paper. The authors also extend thanks to the action editor for the expert handling of the manuscript and to the anonymous reviewers for their valuable suggestions to improve the paper.

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

# APPENDIX

## A   Assumptions

We begin by revisiting the theoretical assumptions listed in the main paper:

- The observed data $\{(X_t, Y_t)\}_{t=1}^n$ is strictly stationary, non-constant, and the underlying distribution $F_{XY_{-l}}$ has finite moments for any lag $l \geq 0$.

- There exists a maximum dependence lag $M$ such that for all $l \geq M$, the two time series are almost independent for large $n$, so are each time series within itself:

$$\sup|F_{XY_{-l}} - F_X F_Y| = O(\frac{1}{n}),$$

$$\sup|F_{XX_{-l}} - F_X F_X| = O(\frac{1}{n}),$$

$$\sup|F_{YY_{-l}} - F_Y F_Y| = O(\frac{1}{n}).$$

- The maximum dependence lag $M$ and the maximum lag under consideration $L$ are non-negative integers that satisfies $L \geq M$ and $L = o(n)$, i.e., they may increase together with $n$ but at a slower pace.

- As the sample size $n$ increases, both the number of blocks $B$ and the number of observations per block $\frac{n}{B}$ increase to infinity. Moreover, $\frac{n}{B} \geq M$ for sufficiently large $n$.

- The sample dependence measure has the following form:

$$\tau_n(\vec{X}, \vec{Y}) = \frac{\sum_{i=1}^n \sum_{j=1}^n \gamma_n(i,j)}{n^2},$$

where each $\gamma_n(i,j)$ is a function of $(X_i, X_j, Y_i, Y_j)$, and remaining sample pairs may also be used but with a weight of $O(1/n)$.

- In the standard i.i.d. setting where $(X_1, Y_1), (X_2, Y_2), \ldots, (X_n, Y_n) \overset{i.i.d.}{\sim} F_{XY}$, there exists a population statistic $\tau(X, Y)$ defined solely based on the joint distribution $F_{XY}$. Each term in the sample statistic satisfies:

$$\mathbb{E}(\gamma_n(i,j)) = \tau(X, Y) + o(1).$$

Moreover, the population statistic $\tau(X, Y)$ is non-negative and equals 0 if and only if $X$ and $Y$ are independent, i.e., $F_{XY} = F_X F_Y$.

## B   Theorem Proofs

**Theorem 1.** *The cross dependence sample statistic satisfies:*

$$EE(\tau_n(\vec{X}, \vec{Y}_{-l})) - \tau(X, Y_{-l}) = o(1),$$

$$Var(\tau_n(\vec{X}, \vec{Y}_{-l})) = O(\frac{1}{n-l}).$$

*Therefore, for each $l \in \{0, ..., L\}$, we have*

$$\tau_n(\vec{X}, \vec{Y}_{-l}) \overset{n \to \infty}{\Rightarrow} \tau(X, Y_{-l})$$

*in probability.*

*Proof.* First, applying the assumptions on the dependence measure to the cross dependence statistics yields:

$$\tau_n(\vec{X}, \vec{Y}_{-l}) = \frac{\sum_{i=l+1}^{n} \sum_{j=l+1}^{n} \gamma_{n-l}(i,j)}{(n-l)^2},$$

$$\mathbb{E}(\gamma_{n-l}(i,j)) = \tau(X, Y_{-l}) + o(1).$$

Here, each $\gamma_{n-l}(i,j)$ is a function of $(X_i, X_j, Y_{i-l}, Y_{j-l})$, and remaining sample pairs like $(X_u, X_v, Y_w, Y_z)$ may also be used but with a weight of $O(1/n)$.

As expectations are additive, it immediately follows that

$$\begin{aligned}
\mathbb{E}(\tau_n(\vec{X}, \vec{Y}_{-l})) &= \frac{\sum_{i=l+1}^{n} \sum_{j=l+1}^{n} \mathbb{E}(\gamma_{n-l}(i,j))}{(n-l)^2} \\
&= \frac{\sum_{i=l+1}^{n} \sum_{j=l+1}^{n} \{\tau(X, Y_{-l}) + o(1)\}}{(n-l)^2} \\
&= \tau(X, Y_{-l}) + o(1).
\end{aligned}$$

Next, the variance equals

$$\text{Var}(\tau_n(\vec{X}, \vec{Y}_{-l})) = \frac{Cov(\sum_{i=l+1}^{n} \sum_{j=l+1}^{n} \gamma_{n-l}(i,j), \sum_{u=l+1}^{n} \sum_{v=l+1}^{n} \gamma_{n-l}(u,v))}{(n-l)^4}.$$

Therefore, it suffices to consider each covariance term $Cov(\gamma_{n-l}(i,j), \gamma_{n-l}(u,v))$, and there are $(n-l)^4$ such terms.

When both $|u - i| > M$ and $|v - j| > M$, the maximum dependent lag possible, we have

$$Cov(\gamma_{n-l}(i,j), \gamma_{n-l}(u,v)) = O(\frac{1}{n-l}).$$

Otherwise it is

$$Cov(\gamma_{n-l}(i,j), \gamma_{n-l}(u,v)) = O(1).$$

There are a total of $O((n-l)^2(n-M)^2)$ covariance terms of magnitude $O(\frac{1}{n-l})$, while the remaining $O((n-l)^3)$ covariance terms are of magnitude $O(1)$. Consequently, as $M = o(n)$, we have

$$\begin{aligned}
\text{Var}(\tau_n(\vec{X}, \vec{Y}_{-l})) &= \frac{O((n-l)^2(n-M)^2) * O(\frac{1}{n-l}) + O((n-l)^3) * O(1)}{(n-l)^4} \\
&= O(\frac{1}{n-l}),
\end{aligned}$$

which converges to 0 as $n$ increases.

With the expectation converging to the population statistic and the variance approaching 0, we can conclude that

$$\tau_n(\vec{X}, \vec{Y}_{-l}) \overset{n \to \infty}{\Rightarrow} \tau(X, Y_{-l})$$

in probability. □

**Theorem 2.** *The temporal dependence sample statistic satisfies:*

$$\text{T}_n(\vec{X}, \vec{Y}) \overset{n \to \infty}{\Rightarrow} \sum_{l=0}^{L} \tau(X, Y_{-l}).$$

*The estimated optimal dependence lag satisfies:*

$$\hat{L}^* \overset{n \to \infty}{\Rightarrow} \arg \max_{l \in [0,L]} \tau(X, Y_{-l}).$$

*Proof.* By Theorem 1, each $\tau_n(\vec{X}, \vec{Y}_{-l})$ converges to $\tau(X, Y_{-l})$ with a variance of $O(\frac{1}{n-l})$.

Recall the definition of the temporal dependence statistic $\mathrm{T}_n(\vec{X}, \vec{Y})$ as

$$\mathrm{T}_n(\vec{X}, \vec{Y}) = \sum_{l=0}^{L} \left( \frac{n-l}{n} \right) \cdot \tau_n(\vec{X}, \vec{Y}_{-l}).$$

Then the expectation satisfies

$$\mathbb{E}(\mathrm{T}_n(\vec{X}, \vec{Y})) = \sum_{l=0}^{L} \left( \frac{n-l}{n} \right) \cdot \tau(X, Y_{-l}) + o(L)$$

$$\stackrel{n\to\infty}{\Rightarrow} \sum_{l=0}^{L} \tau_n(\vec{X}, \vec{Y}_{-l})$$

by noting that $L$ is fixed and the weight $\frac{n-l}{n}$ converges to 1. Moreover, the variance satisfies

$$Var(\mathrm{T}_n(\vec{X}, \vec{Y})) = O(\frac{L^2}{n-L}),$$

which also converges to 0. Consequently,

$$\mathrm{T}_n(\vec{X}, \vec{Y}) \stackrel{n\to\infty}{\Rightarrow} \sum_{l=0}^{L} \tau(X, Y_{-l})$$

in probability.

Similarly, the estimated optimal dependence lag satisfies

$$\hat{L}^* = \arg \max_{l \in [0,L]} \left( \frac{n-l}{n} \right) \cdot \tau_n(\vec{X}, \vec{Y}_{-l})$$
$$\stackrel{n\to\infty}{\Rightarrow} \arg \max_{l \in [0,L]} \tau(X, Y_{-l})$$

in probability. □

**Theorem 3** (Asymptotic Validity). *Under the null hypothesis that $\vec{X}$ and $\vec{Y}$ are independent for all lags $l \in [0, L]$, the test statistic satisfies:*

$$\mathrm{T}_n(\vec{X}, \vec{Y}) \stackrel{n\to\infty}{\Rightarrow} 0.$$

*Moreover, the block-permutation test is asymptotically valid, i.e.,*

$$Prob(\mathrm{T}_n(\vec{X}, \vec{Y}) \geq z_{n,\alpha}) \stackrel{n\to\infty}{\Rightarrow} \alpha.$$

*Proof.* By Theorem 2, we have

$$\mathrm{T}_n(\vec{X}, \vec{Y}) \stackrel{n\to\infty}{\Rightarrow} \sum_{l=0}^{L} \tau(X, Y_{-l}).$$

From the assumption of the population measure, when $X_t$ and $Y_t$ are independent for all lags $l \in [0, L]$, we must have

$$\tau(X, Y_{-l}) = 0$$

for all $l \in [0, L]$. As a result,

$$\mathrm{T}_n(\vec{X}, \vec{Y}) \stackrel{n\to\infty}{\Rightarrow} 0$$

To establish the asymptotic validity of the block permutation test, it suffices to prove that when $\vec{X}$ and $\vec{Y}$ are independent, we have:

$$\sup|F_{T_n(\vec{X},\vec{Y})} - F_{T_n^b}| \overset{n\to\infty}{\to} 0.$$

In other words, if the true null distribution and the permuted distribution is asymptotically the same, then it follows that under the null hypothesis:

$$Prob(T_n(\vec{X},\vec{Y}) \geq z_{n,\alpha}) \overset{n\to\infty}{\to} \alpha.$$

Here, $T_n(\vec{X},\vec{Y})$ is a function of $(X_i, X_j, Y_u, Y_v)$ for $i, j, u, v = 1, 2, \ldots, n$, and the permuted statistic $T_n(\vec{X}, \vec{Y}_{\pi_B})$ is the same function but on $(X_i, X_j, Y_{u'}, Y_{v'})$, where $u'$ and $v'$ represent the permuted indices of $u$ and $v$. Therefore, it suffices to prove that under the null hypothesis, the distribution of $(X_i, X_j, Y_{u'}, Y_{v'})$ converges to the distribution of $(X_i, X_j, Y_u, Y_v)$ for sufficiently large $n$. Note that under the standard i.i.d. setting, these two distributions are identical under the null hypothesis where $X$ and $Y$ are independent.

We first consider the case where both $u$ and $v$ belong to the same block. In this case, $u'$ and $v'$ will also be in the same block and differ by the same lag difference. Furthermore, due to the stationary assumption, $F_{Y_u,Y_v} = F_{Y_{u'},Y_{v'}}$. Now, as we are examining the null distribution where $\vec{X}$ and $\vec{Y}$ are independent, it follows that

$$F_{X_i,X_j,Y_u,Y_v} = F_{X_i,X_j}F_{Y_u,Y_v} = F_{X_i,X_j}F_{Y_{u'},Y_{v'}} = F_{X_i,X_j,Y_{u'},Y_{v'}}.$$

Namely, $(X_i, X_j, Y_u, Y_v)$ and $(X_i, X_j, Y_{u'}, Y_{v'})$ are identically distributed in this case.

Next we examine the case where $u$ and $v$ belong to different blocks. Given our assumption of a maximum dependence lag $M$, if $|u - v| > M$ and $|u' - v'| > M$ for the permuted indices, we can establish the following:

$$\begin{aligned}
&\sup|F_{X_i,X_j,Y_u,Y_v} - F_{X_i,X_j,Y_{u'},Y_{v'}}| \\
=&\sup|F_{X_i,X_j}F_{Y_u,Y_v} - F_{X_i,X_j}F_{Y_{u'},Y_{v'}}| \\
\leq&\sup|F_{X_i,X_j}(F_{Y_u,Y_v} - F_{Y_u}F_{Y_v})| + \sup|F_{X_i,X_j}(F_{Y_u}F_{Y_v} - F_{Y_{u'}}F_{Y_{v'}})| \\
&+ \sup|F_{X_i,X_j}(F_{Y_{u'}}F_{Y_{v'}} - F_{Y_{u'},Y_{v'}})| \\
=& \, o(1)
\end{aligned}$$

Here, the first and third terms are $o(1)$ as per our maximum dependence lag assumption, while the second term is exactly 0 because the marginals within the brackets remain identical before and after permutation. Consequently, in this case, $(X_i, X_j, Y_u, Y_v)$ is asymptotically equivalent in distribution to $(X_i, X_j, Y_{u'}, Y_{v'})$.

Finally, in the case where $u$ and $v$ belong to different blocks, there are two additional possibilities: either $|u - v| \leq M$ or $|u - v| \leq M$. In either case, we no longer have exact distribution equivalence nor asymptotic equivalence. The number of instances where $(X_i, X_j, Y_u, Y_v)$ does not match $(X_i, X_j, Y_{u'}, Y_{v'})$ in distribution is at most $O(MB)$, which equals $o(n^2)$ by our assumption on $M$ and $B$.

Therefore, taking all the above arguments together, as the sample size $n$ goes to infinity, we have:

$$Prob(\sup|F_{X_i,X_j,Y_u,Y_v} - F_{X_i,X_j,Y_{u'},Y_{v'}}| \to 0) \to 1$$

for any random block permutation $\pi_B$ satisfying our assumption. As the result,

$$Prob(\sup|F_{T_n(\vec{X},\vec{Y})} - F_{T_n(\vec{X},\vec{Y}_{\pi_B})}| \to 0) \to 1.$$

Namely, the sample statistic and the block-permuted statistic have asymptotically the same distribution, and the test is asymptotically valid. □

**Theorem 4** (Testing Consistency). *Under the alternative hypothesis that $\vec{X}$ and $\vec{Y}$ are dependent for some lag $l \in [0, L]$, the test statistic satisfies*

$$T_n(\vec{X},\vec{Y}) \overset{n\to\infty}{\to} c > 0.$$

*Moreover, the block-permutation test is asymptotically consistent, i.e.,*

$$Prob(T_n(\vec{X},\vec{Y}) \geq z_{n,\alpha}) \overset{n\to\infty}{\to} 1.$$

*Proof.* From the assumption on the dependence measure, when there exists at least one lag $l$ such that the two time series are dependent, we must have:

$$\tau(X, Y_{-l}) = c_{-l} > 0.$$

As all other cross dependence sample statistics are non-negative, it follows that

$$\mathrm{T}_n(\vec{X}, \vec{Y}) \overset{n \to \infty}{\Rightarrow} \sum_{l=0}^{L} c_{-l} > 0$$

To prove consistency under the permutation test, it suffices to show that at any type 1 error level $\alpha$, when the two time series are dependent for some lag, the p-value of sample dependence measure is less than $\alpha$ as the sample size approaches infinity. Note that Theorem 8 in Shen et al. (2020) proved consistency of standard permutation test between two i.i.d. sample data, and the follow-on proof has similar steps but with significant adjustment for the block permutation procedure.

In the block permutation test, the p-value can be expressed as follows:

$$Prob(\mathrm{T}_n(\vec{X}, \vec{Y}_{\pi_B}) > \mathrm{T}_n(\vec{X}, \vec{Y}))$$
$$= \sum_{w=0}^{B} Prob(\mathrm{T}_n(\vec{X}, \vec{Y}_{\pi_B}) > \mathrm{T}_n(\vec{X}, \vec{Y}) | \pi_B \text{ is a partial derangement of size } w)$$
$$\times Prob(\text{partial derangement of size } w).$$

This expression conditions on the block permutation being a partial derangement of size $w \in [0, B]$, where $w = 0$ implies that $\pi_B$ is a derangement where no two blocks remain in their original positions, and $w = B$ means that $\pi_B$ does not permute any blocks.

As $B \to \infty$, from the basic property of derangement[4] we have

$$Prob(\text{partial derangement of size } w) \to e^{-1}/w!.$$

Because $\mathrm{T}_n(\vec{X}, \vec{Y}) \to c > 0$ under dependence, it suffices to prove that for any $c > 0$,

$$\lim_{n \to \infty} e^{-1} \sum_{w=0}^{B} Prob(\mathrm{T}_n(\vec{X}, \vec{Y}_{\pi_B}) > c | \text{ partial derangement of size } w)/w! \to 0. \tag{1}$$

We decompose the above summations into two different cases. The first case is when $w$ is of fixed size, then $\vec{X}$ and $\vec{Y}_{\pi_B}$ are asymptotically independent. This is because, for fixed $w$, the number of observations that are not moved is fixed and asymptotically goes to 0, and all remaining blocks are shifted to different positions. By the maximum dependence lag $M$, which is $o(n)$, and the number of samples per block being larger than $M$, the block permutation makes all other observation pairs asymptotically independent. Therefore, given $i, j$, and $i', j'$ being their block-permuted indices, we must have

$$\sup|F_{X_i, X_j, Y_{i'}, Y_{j'}} - F_{X_i, X_j} F_{Y_{i'}, Y_{j'}}| = o(1)$$

so long $|i' - i| > M$ and $|j' - j| > M$, which asymptotically holds for all blocks who moved the block position. Therefore, when $w$ is a fixed size, $\vec{X}$ and $\vec{Y}_{\pi_B}$ are asymptotically independent, and we have

$$\mathrm{T}_n(\vec{X}, \vec{Y}_{\pi_B}) \to 0$$

as the sample statistic converges to the population, and the population statistic equals 0 under independence.

The other case is the remaining partial derangements $\pi_B$ of increasing size $w$, but these partial derangements occur with probability converging to 0. Formally, for any $\alpha > 0$, there exists $B_1$ such that

$$e^{-1} \sum_{w=B_1+1}^{+\infty} 1/w! < \alpha/2.$$

---

[4] https://en.wikipedia.org/wiki/Rencontres_numbers

This is because $\sum_{w=0}^{B} 1/w!$ is bounded above and converges to $e$. Then back to the first case, we can find $B_2 > B_1$ such that for any $w \le B_1$ and all $B > B_2$,

$$Prob(\mathrm{T}_n(\vec{X}, \vec{Y}_{\pi_B}) > c|\text{ partial derangement of size } w) < \alpha/2.$$

Therefore, for all $B > B_2$:

$$e^{-1} \sum_{w=0}^{B} Prob(\mathrm{T}_n(\vec{X}, \vec{Y}_{\pi_B}) > c|\text{ partial derangement of size } w)/w!$$

$$< e^{-1} \sum_{w=0}^{B_1} \alpha/2w! + e^{-1} \sum_{w=B_1+1}^{B} 1/w!$$

$$< \alpha.$$

Thus the convergence in Equation 1 holds.

In conclusion, at any type 1 error level $\alpha > 0$, the p-value of the temporal dependence sample statistic under the block permutation test will eventually be less than $\alpha$ as $n$ increases. Therefore, the proposed test is consistent against all dependencies with finite second moments, and its testing power converges to 1 when the time series $\vec{X}$ and $\vec{Y}$ are dependent. $\qquad\square$

