# OpenReview forum: "Independence Testing for Temporal Data"
_TMLR — Accepted by TMLR_

### Review · Reviewer_WM9r · 2024-03-12

**Summary Of Contributions:**

This paper proposes a simple but powerful independence test for structured data, with clearly stated assumptions, good theoretical guarantees, and corroborative experiments.

**Audience:**

Yes

**Claims And Evidence:**

Yes

**Requested Changes:**

*Critical*:
- None!

*Suggested*: (in order of appearance)
- be consistent with either "time-series" or preferably "time series"
- "... can inflate the p-value and result in [an] invalid test."
- "... fMRI data reveals various temporal [dependences]..."
- add discussion of xi correlation/Chatterjee's coefficient (e.g., take a look at Azadkia and Chatterjee (2021) and Shi, Drton, Han (2021))
- "criterion" instead of "Criterion" when writing out HSIC
- "The proposed method consists [of] four steps:"
- consistently use  "Type I error" instead of "type 1" or "type-1"
- write "number of blocks" instead of "block size"
- remove extra ")" in third sentence of Section 3.3
- be consistent with "p-value" vs "*p*-value"

**Strengths And Weaknesses:**

*Strengths*:
- explicit assumptions
- simple method
- clear presentation
- strong theoretical results

*Weaknesses*:
- some terminology/notation inconsistencies and typos

---

> ### Author Response · Authors · 2024-04-01
>
> Thank you for reviewing our manuscript. We have corrected all suggested minor issues.
>
> Regarding Chatterjee's coefficient, in Section 2.3 (page 3) of the revised paper, we discussed it alongside several other recent dependence measures and why they are all compatible with the testing process.
>
> The paper addition (citations were removed in the response, but are in the revised paper) are copied below:
>
> "As the proposed temporal statistic is essentially an aggregation of the underlying dependence measure, its effectiveness in capturing dependence is contingent upon the choice of dependence measure. It is well known that each dependence measure has its own unique strengths. Therefore, our usage of distance and kernel based statistics in this paper should be viewed as an illustration of the validity and consistency properties of the proposed temporal test.
>
> Some examples of other dependence measures include correlation coefficients, Chatterjee’s rank correlation, the HHG method, projection correlation, ball covariance, as well as recent high-dimensional dependence statistics. All of these dependence measures can be directly incorporated into our temporal testing framework by simply modifying the cross-dependence statistics in Step 1. Such adaptations may offer better testing power for certain dependence structures.
>
> For instance, using the correlation coefficient with block permutation will only detect linear associations in temporal data, while a universally consistent dependence measure can detect all possible dependencies with a sufficiently large sample size; dependence measures that are better at detecting nonlinear or high-dimensional dependencies in standard i.i.d.~data will also perform better under such dependencies in the case of temporal data, requiring a smaller sample size to achieve perfect testing power; rank-based dependence measures can be more robust against data noise. "

---

### Review · Reviewer_fn2g · 2024-03-14

**Summary Of Contributions:**

The authors present an approach to statistical testing of independence for time series data. Building on previous work on non-parametric independence testing, the proposed approach consists of a sequence of steps which themselves have been well established, but in their composition would be interesting as a simple and modular approach to time series testing. The theoretical part of the work could be discussed better in light of the existing literature, in its present form the manuscript seems to have a rather small focus. As for the experimental evaluation there seems to be a bit of a gap betwen the rather simple set of simulation studies (univariate, very simple temporal dependency, no autocorrelation structure) followed by an experiment on fMRI data  (multivariate, very complex spatiotemporal dependencies, strong autocorrelation structure).

**Audience:**

Yes

**Broader Impact Concerns:**

Testing for independence is an important tool that could be used in a variety of scenarios once published in a standard package. The impact of methods that do not correctly identify independence could be broad, hence assessment under realistic conditions would be important.

**Claims And Evidence:**

No

**Requested Changes:**

For a better assessment of the pros and cons of the proposed method, it would be great if the authors could:

* embed the work a bit better in existing literature and discuss connections with other approaches / theoretical work
* work on the experimental evaluation:
    * try to close the gap between simulation and real world data by considering multivariate data that has similar characteristics as the fMRI data, more realistic noise and autocorrelation structure
    * maybe compare with existing methods for the same application scenario

**Strengths And Weaknesses:**

## Strengths
* Testing independence for time series is a relevant and interesting topic
* The experimental evaluation included several synthetic and one real world data set
* Block permutations could be a feasible approach to independence testing in this scenario
* The proposed algorithm is intuitive and simple. It can leverage different dependence statistics.

## Weaknesses
* *Non-stationarity assumption*: One of the main assumptions of the proposed algorithm is stationarity of the data. Almost all other algorithms in the statistical learning literature rest on that assumption, so it's a standard assumption. But in this particular setting, time series with potentially slow frequencies (in particular frequencies slower than the Nyquist frequency of the signals in the data), this assumption could deserve some more attention. If there were trends / slow oscillations in the data, the stationarity between the blocks could be violated; or at least these variations in the data could affect the dependence measures. Often time series methods employ some preprocessing, such as temporal difference operators, to reduce these effects. I would assume that in some cases similar techniques could make sense for the proposed method as well?
* *Lag estimation*: An important step in the proposed algorithm is the identification of the correct lag. The size of the blocks and the autocorrelation structure of the data are intimately related to this estimation process. It would be great to see how the proposed approach performs in more realistic and controlled conditions, where there are multivariate time series with more complex temporal dynamics governing the dependency structure between the two time series.
* *Experimental evaluation*: The experimental evaluation is well done in some parts, and I do appreciate the synthetic data experiments, which are simple but highlight some main aspects. However there seems to be a substantial gap between the synthetic data experiments (univariate, very simple temporal dependency, no autocorrelation structure) and the real world data (multivariate, very complex spatiotemporal dependencies, strong autocorrelation structure). Especially for time series data with few samples and very high dimensionality, the proposed methods should be very robust to noise and confounding factors in the autocorrelation structure.
* *Related work*: While the work appears interesting, there seems to be several fields of research relevant to this work. I don't think it's necessary to reference all of these below, but a quick literature research surfaced a number of scientific communities that have worked on the very problem discussed in the present study from a theoretical side as well as on the application side (fMRI data):

    * **ICA**: There appears to be a a lot for work in the ML community around independence of time series in the Independent Component Analysis (ICA) community (see e.g. [Independent component analysis: algorithms and applications](https://www.sciencedirect.com/science/article/abs/pii/S0893608000000265)). Some other factor model based approaches also use the idea of temporal embeddings, which is similar to the lagged time series approach in the present study [Temporal Kernel CCA](https://link.springer.com/article/10.1007/s10994-009-5153-3)
    * **Mixing**: The notion of mixing processes has been investigated for some time in the statistics community - these references are not meant to be representative, I just picked some that appeared relevant top the application scenario and that highlight the lines of work: [SOME MIXING PROPERTIES OF TIME SERIES MODELS](https://d1wqtxts1xzle7.cloudfront.net/69497400/0304-4149_2885_2990031-620210912-26917-167rbdn-libre.pdf?1631501125=&response-content-disposition=inline%3B+filename%3DSome_mixing_properties_of_time_series_mo.pdf&Expires=1710414657&Signature=aJgJw5OqipMAnhUSm9SgTGGHbK2pUCEt9Aann6JEh2fNofSs57~6louo6n3aknFiHRo6~8SlMXbFJxI1-YSiCCtz~wSWdJpSAG2xrke2UfGK8ayU1h~Yh2MpUZsb5GycQxh6Kd0TOiKBuaI7d8jHun8Ub5xednO3H1WSlURBbcKa4j2h7DkjiTMoIR8f9BOXJQA6oaRvD85-afY9jbikFAiGPZPrCuL1ID5Zesx~Omok3KrHEOWnc~9EdllV1KqrT3TerMn-ZlM73sNBhXeXsIGXtKTnblNjdtzpYFv~81j~Jx1iXJvAhs~tiErPn4deVQ5uVKFlyz~UlxtzSOx0FA__&Key-Pair-Id=APKAJLOHF5GGSLRBV4ZA), [Estimating beta-mixing coefficients](https://proceedings.mlr.press/v15/mcdonald11a.html), [Learning with little mixing](https://proceedings.neurips.cc/paper_files/paper/2022/hash/1dc9fbdb6b4d9955ad377cb983232c9f-Abstract-Conference.html), [A Note on Mixing in High Dimensional Time Series](https://arxiv.org/abs/1911.10648)
    * **Causality** The methods developed in econometrics and applied to neuroimaging data appear to be relevant to the problem tackled here, too. My understanding was that some of these often used methods don't really work on real world neuroimaging data, mostly due to low specificity / testing power and high noise of neuroimaging data, see also [A critical assessment of connectivity measures for EEG data: A simulation study](https://www.sciencedirect.com/science/article/abs/pii/S1053811912009469), [Sparse causal discovery in multivariate time series](https://proceedings.mlr.press/v6/haufe10a), [Validity of Time Reversal for Testing Granger Causality](https://ieeexplore..ieee.org/abstract/document/7412766)
    * **ICA for neuroimaging/fMRI** There seems to be a substantial body of literature on ICA for fMRI, which could be discussed or at least compared against, see e.g. [Blind Source Separation of Multiple Signal Sources of fMRI Data Sets Using Independent Component Analysis](https://journals.lww.com/jcat/fulltext/1999/03000/blind_source_separation_of_multiple_signal_sources.16.aspx) or [Validating the independent components of neuroimaging time series via clustering and visualization](https://www.sciencedirect.com/science/article/pii/S1053811904001661)

---

> ### Author Response · Authors · 2024-04-01
> **On multivariate simulation (for lag estimation and experimental evaluation)**
>
> Thank you very much for the suggestion regarding lag estimation and experimental evaluation.
>
> We have included a new section 4.3 on page 9 for multivariate simulations, featuring testing power in Figure 5 (page 11) and optimal lag estimation in Figure 6 (page 12). To make them more interesting, we maintained a fixed sample size $n=100$ while increasing dimensionality, and designed a decaying weight matrix $D$, for a meaningful multivariate simulation where additional dimensions contain weaker dependence signals.
>
> Despite the anticipated decrease in power / accuracy as dimensionality increases, our proposed method consistently outperforms the competitor method in testing power, and demonstrates satisfactory performance in lag estimation.
>
> Note that in the multivariate simulation, if the dimension is fixed, the testing power eventually reaches 1 due to the consistency property, which is noticeable in the case of small $p$. For example, in case of multivariate linear, the testing power is about $0.8$ at $n=100$ and $p=20$. In fact, if we had increased the sample size $n$ to be larger, it would require higher dimensionality to diminish the power.

---

> ### Author Response · Authors · 2024-04-01
> **On ICA**
>
> Thank you for the suggestion regarding ICA, which is indeed commonly used in multivariate time series analysis. However, ICA is a dimension reduction technique designed for a single multivariate time series. It identifies independent components that remove dependence and correlations within the data. Therefore, as a dimension reduction technique, it is not directly applicable to dependence testing between two time-series.
>
> However, if we consider the problem from a reversed perspective, dependence measures can serve as effective tools for dimension reduction, with feature screening being a popular technique in this regard. As such, we have included the following in the conclusion (page 13) to highlight this connection:
>
> "Secondly, dependence measures are commonly employed in dimension reduction techniques, such as screening, especially in high-dimensional data settings. However, little attention has been given to the temporal domain. While it is straightforward to utilize dependence measures for dimension reduction in multivariate time series, delving into their theoretical properties and their relationships with other standard tools, such as independence component analysis, could provide valuable insights."

---

> ### Author Response · Authors · 2024-04-01
> **On Non-stationary, Mixing, and Causality**
>
> Thank you for all the excellent suggestions. We have incorporated additional text and provided justification for each of them in the revised paper. Below are the details:
>
> (Note: The paper changes include numerous additional citations, including those suggested in your review, which are referenced as \citep below.)
>
> 1. Non-Stationary:
>
> It is a common practice to process non-stationary time series into stationary ones before applying further statistical analysis. In Section 3.1 on page 5, we have included a brief overview of these techniques, thereby providing justification for the stationary assumption.
>
> "For non-stationary data, there exist many common techniques to remove trends and process them into approximately stationary processes \citep{cleveland1990stl,hastie2009elements,enders2010applied,shumway2010time,box2015time}. Some examples include differencing, where one computes the difference between consecutive observations; detrending via linear regression or polynomial fitting and subtracting the trend component from the original series; seasonal adjustment by decomposition; log / square root / Box-Cox transformation to stabilize variance; smoothing via moving averages to reduce noise and short-term fluctuations; filtering to remove specific frequencies from the data."
>
> 2. Mixing:
>
> A stochastic process is mixing if its values at widely-separated times are asymptotically independent. Therefore, this is akin to our assumption that there exists a maximum lag beyond which the time series components are independent. Consequently, our results can be viewed as approximately true for mixing time series as well. We opt to approach the theoretical results using the maximum lag, as it facilitates the theorem proofs and also aligns nicely with optimal lag estimation.
>
> In Section 3.1 on page 5, we add the following explanation after the maximum lag assumption:
>
> "The second and third assumptions require that the time series exhibit independence for sufficiently large lags beyond $M$, and that the maximum lag to be examined, $L$, must be no less than $M$. Such an assumption shares similarity with the mixing property, where a stochastic process is mixing if its values at widely-separated times are asymptotically independent \citep{Tran1985,McDonald11a,Ziemann2022}. Therefore, our results can be viewed as valid for mixing time series as well."
>
> 3. Causality:
>
> Indeed, causal inference in time series data is highly desirable. That said, this topic is not directly related to our paper, as correlation does not imply causality. However, there are some recent works utilizing dependence measures for causality. To that end, we have added the following paragraph to the conclusion on page 13 to acknowledge the connection and identify avenues for future research:
>
> "Thirdly, causal inference in time series data is an important task \citep{haufe2010sparse,winkler2016validity}. While it is widely recognized that correlation does not imply causality, recent research has demonstrated the utility of dependence and conditional dependence tests in causal inference \citep{cai2022distribution,laumann2023kernel}. Therefore, extending this framework to encompass conditional independence and causal inference may significantly advance the understanding of causal inference in time series data."

---

### Review · Reviewer_xdHS · 2024-03-20

**Summary Of Contributions:**

This paper proposes a new permutation test for independence between two (multi-variate) time series. Since the test is permutation-based, various existing correlation measures can be used for testing. Theoretical justifications as well as numerical examination of the proposed method are given, and the results are interesting.

**Audience:**

Yes

**Broader Impact Concerns:**

No concern here.

**Claims And Evidence:**

Yes

**Requested Changes:**

1. I notice that although the methodology is described for multi-variate time series, all examples given in the simulation study and real data analysis are bivariate. It would also be interesting to show how the testing power changes as the dimension increases in the simulation study. For the real data analysis, one can also test for independence between different networks, in addition to ROIs.

2. I have some doubts about the proof of Theorem~3, where the authors state that "it suffices to prove that under the null hypothesis, $(X_i, X_j, Y_u, Y_v)$ and $(X_i, X_j, Y_{u′}, Y_{v′} )$ are identically distributed ". However, the randomness of the permutation test statistic comes from the random permutation while conditioned on the observed data. I don't think such an argument will hold. Perhaps the proof can be modified similarly to those in [1], for example.

References:

[1] DiCiccio, C. J., & Romano, J. P. (2017). Robust permutation tests for correlation and regression coefficients. Journal of the American Statistical Association, 112(519), 1211-1220.

**Strengths And Weaknesses:**

Strengths: the paper is well-written and easy to follow. The proposed method is a welcomed addition to existing literature.
Weaknesses: Some theoretical proofs may need to be modified.

---

> ### Author Response · Authors · 2024-04-01
> **On multivariate simulation**
>
> Thank you very much for the suggestion regarding multivariate simulations!
>
> We have included a new section 4.3 on page 9 for multivariate simulations, featuring testing power in Figure 5 (page 11) and optimal lag estimation in Figure 6 (page 12). To make them more interesting, we maintained a fixed sample size while increasing dimensionality, and designed a decaying weight matrix $D$, for a meaningful multivariate simulation where additional dimensions contain weaker dependence signals.
>
> Despite the anticipated decrease in power / accuracy as dimensionality increases, our proposed method consistently outperforms the competitor method in testing power, and demonstrates satisfactory performance in lag estimation.
>
> Note that in the multivariate simulation, if the dimension is fixed, the testing power eventually reaches 1 due to the consistency property, which is noticeable in the case of small $p$. For example, in case of multivariate linear, the testing power is about $0.8$ at $n=100$ and $p=20$. In fact, if we had increased the sample size $n$ to be larger, it would require higher dimensionality to diminish the power.

---

> ### Author Response · Authors · 2024-04-01
> **On the proof of Theorem 3**
>
> Thank you for bringing this valuable paper to our attention. We have added the citation to the paper as additional evidence that the standard permutation test does not apply to correlated data.
>
> We also appreciate your careful review of our proof and the concern you raised. Upon a careful examination of the proof steps, we believe that the overall argument remains valid. Specifically, the theorem is about asymptotic validity, and we were trying to show they are asymptotically the same in distribution, which indeed is not exactly the same in the temporal setting.
>
> Regarding the sentence in question, we have clarified it as follows:
>
> "Therefore, it suffices to prove that under the null hypothesis, the distribution of $(X_i,X_j, Y_{u'},Y_{v'})$ converges to the distribution of $(X_i,X_j, Y_u,Y_v)$ for sufficiently large $n$. Note that under the standard i.i.d.setting, these two distributions are identical under the null hypothesis where $X$ and $Y$ are independent."
>
> We also refined our proof logic in the updated proof. Here is a simplified and more verbal version of our proof logic:
>
> An independence test is valid if and only if the original statistic and the permuted statistic have the same distribution under the independence hypothesis. Then, at any type 1 error level $\alpha$, because these two statistics share the same distribution under the null, the testing power shall always equal $\alpha$, rendering the test valid.
>
> The original statistic is a function of $(X_i,X_j, Y_u,Y_v)$, while the permuted statistic is the same function but of $(X_i,X_j, Y_{u'},Y_{v'})$, where $u'$ and $v'$ are permuted indices of $u$ and $v$. Thus, if $(X_i,X_j, Y_u,Y_v)$ and $(X_i,X_j, Y_{u'},Y_{v'})$ have the same distribution, the original and permuted statistics must also have the same distribution.
>
> In the standard i.i.d.setting, $X_i$ and $X_j$ are independent, so are $Y_u$ and $Y_v$, for every $i \neq j$ and $u \neq v$. Under the null hypothesis where $X$ and $Y$ are independent, the density of $(X_i,X_j, Y_u,Y_v)$ is simply the product of marginals, which always equals $(X_i,X_j, Y_{u'},Y_{v'})$. Therefore, the permutation test is always valid in the standard i.i.d.~setting.
>
> Our Theorem 3 proof follows the same path. However, in this case, the distribution equivalence mentioned above is no longer exact. Depending on whether the original and permuted indices are beyond the maximum lag or not (where beyond the maximum dependence lag is equivalent to the standard i.i.d.setting), the two distributions can be exactly the same, or asymptotically the same with $o(1)$ difference, or not the same at all. Nonetheless, the last case is asymptotically of weight $0$, as the maximum dependence lag is a finite number. Therefore, the two distributions are asymptotically the same, and the block permutation test is asymptotically valid, as validated in the numerical experiment.
>
> Please let us know if there are any issues or ambiguities in the explanation provided above.

---

### Comment · Action_Editor_ShXA · 2024-05-08
**Providing the proofs of the theorems as appendixes**

Hi,

Can I please ask the authors to include the proofs for the four theorems appearing in the paper? You can add those proofs as appendixes. Also, reviewer fn2g said it would be good to add an experiment to "try to close the gap between simulation and real world data by considering multivariate data that has similar characteristics as the fMRI data, more realistic noise and autocorrelation structure". But there was no direct reply to their comment. I agree with the reviewer here, the experimental evaluation seems rather short compared to other papers.

A minor comment. The caption of Table 1 says: "This table displays the parcellation information for the parcels used in our analysis. The first column shows the numeric order of the parcels as they appear in the following figures." There are no following figures. Please, change the order in which the Table and the Figures appear or edit this sentence.

The AC for this paper.

---

> ### Author Response · Authors · 2024-05-08
> **Quick clarification questions:**
>
> Thank you for the comments! We will correct the minor typo asap.
>
> Additionally, we would like to ask some quick clarifications regarding the other questions:
>
> 1. The theorem proofs are currently in the supplementary materials. Do you suggest that we move the proofs to the main PDF, as an appendix section at the very end?
>
> 2. Regarding the multivariate simulations, we have indeed revised the paper accordingly: we added section 4.3 (page 9), along with the simulation results in Figures 5 (page 11) and Figure 6 (page 12). As these simulations are generalizations of the 1D simulations (Sections 4.1 & 4.2), the introductions weren't as extensive.
>
> However, we are more than happy to provide more multivariate simulations to further strengthen this subsection, if that is what you were suggesting?

---

> > ### Comment · Action_Editor_ShXA · 2024-05-09
> >
> > Thanks.
> >
> > 1. Can you update the manuscript and link the theorems to their proofs in the supplemental material? E.g. A comment along the lines of "The proof appears in the supplemental material, appendix XXX".
> > 2. According to the reviewer's comment, please add an experiment with a real dataset, along the lines of Section 5.

---

### Decision · Action_Editor_ShXA · 2024-05-24

**Recommendation:** Accept as is

**Comment:**

The findings in the paper are backed up both by theoretical developments and a diverse of empirical results. This area of research is important in many applied domains and has connections to many subfields of machine learning.

**Audience:**

The problem addressed in this paper is very general and the approach taken here will be of interest to a large community of researchers interested in statistical testing for dependency among time-series data.

**Claims And Evidence:**

Claims are supported by theory and experiments.